# The Role of Patient Expectations in Treatment Outcome and Satisfaction in Osteoarthritis: A Scoping and Mapping Review

**DOI:** 10.3390/jcm14238440

**Published:** 2025-11-28

**Authors:** Mar Flores-Cortés, Tasha R. Stanton, Ferran Cuenca-Martínez

**Affiliations:** 1Faculty of Health Sciences, University of Málaga, 29010 Málaga, Spain; 2IIMPACT in Health, Allied Health and Human Performance, University of South Australia, G.P.O. Box 2471, Adelaide 5001, Australia; 3Biopsychosocial Health Research Group (BPS-Health), Department of Physiotherapy, University of Valencia, c/ Gascó Oliag n°5, 46010 Valencia, Spain

**Keywords:** osteoarthritis, recovery, satisfaction, patient expectations, self-efficacy, pain, scoping review

## Abstract

Recovery in osteoarthritis (OA) is a multidimensional construct that extends beyond symptom reduction to encompass how individuals make sense of, adapt to, and live with their condition. Patient expectations appear to play a central role in shaping this process, influencing how individuals define recovery, cope with functional limitations, and evaluate treatment outcomes. Understanding these expectations is essential for improving satisfaction and functional progress in OA management. This scoping review synthesized evidence on patient expectations in knee, hip and generalized OA and examined how these expectations influence treatment outcome and satisfaction. A PRISMA-ScR–informed scoping review was conducted across PubMed, Embase, CINAHL, PsycINFO, and Web of Science up to September 2025. Eligible studies included adults with OA reporting on treatment expectations, expectation fulfillment, or expectations about their ability to cope (self-efficacy). Both qualitative and quantitative designs were included. Data were extracted and organized for study characteristics, population, expectation domains, and outcomes. Sixteen studies met the inclusion criteria, encompassing qualitative syntheses, cohort studies and systematic reviews. Across study designs, higher or fulfilled expectations were consistently associated with greater satisfaction, pain reduction, and functional improvement. Unrealistic or unfulfilled expectations predicted poorer clinical outcomes and dissatisfaction. Expectations surrounding treatment and clinical outcomes may plausibly influence recovery by shaping physiological responses, emotional reactions and patient engagement with rehabilitation. These findings highlight the importance of aligning clinical interventions with patient expectations. Integrating expectation assessment and recalibration into OA care may enhance adherence and patient satisfaction. Future research should further explore expectations as modifiable therapeutic targets and examine how they interact with factors such as symptom duration, severity and pain extension. Such combinations may clarify which patient profiles are most responsive to expectation-focused interventions. Overall, expectations emerge as a central component of recovery and a promising avenue for more person-centered physiotherapy and rehabilitation practice.

## 1. Introduction

Osteoarthritis (OA) is the most common form of arthritis and one of the leading causes of pain and disability worldwide, affecting more than 300 million people globally [1]. Its prevalence is rising due to aging populations, obesity, and lifestyle factors, contributing substantially to years lived with disability and socioeconomic burden [2,3]. OA is characterized by joint pain, stiffness, and progressive functional limitation, but the lived experience extends far beyond structural changes. Patients frequently report challenges in mobility, work participation, social engagement, and psychological wellbeing [4]. This underscores the need for care models that move beyond a narrow biomedical focus to embrace a person-centered perspective.

Contemporary guidelines from OARSI, ESCEO, AAOS, and NICE consistently emphasize multimodal, non-surgical, and patient-centered strategies for OA management, including exercise, education, weight management, and psychosocial interventions [5,6,7,8]. Despite its multidimensional nature, “recovery” in clinical practice is still often interpreted narrowly and assessed primarily through radiographic findings or pain intensity scales. This symptom-focused approach fails to capture what patients themselves consider meaningful recovery. For many, recovery means being able to resume meaningful roles, restore independence, and re-establish a sense of normality, even if pain persists [9,10].

Patient expectations about treatments are central to this discussion. They shape patients’ willingness to initiate treatments, adherence to rehabilitation, and ultimate satisfaction with care [11]. In surgical contexts such as knee or hip arthroplasty, preoperative expectations are among the strongest predictors of postoperative satisfaction, functional outcomes, and decisional regret [12,13,14]. In conservative management, patients who hold positive yet realistic expectations regarding exercise or self-management demonstrate greater adherence and clinically relevant improvements [15]. Conversely, unrealistic or unmet expectations often result in dissatisfaction, even when objective measures improve [16].

Expectations surrounding treatment and clinical outcome may plausibly influence recovery by affecting physiological responses, emotional reactions, and as well as the ways in which patients engage with their environment and treatment. From a cognitive-affective perspective, expectations influence motivation, engagement, and coping strategies [17]. Neurobiological research on placebo and nocebo responses demonstrates that expectancy and communication framing modulate pain perception via endogenous opioid and dopaminergic pathways [18,19]. Thus, expectations operate not simply as ideas patients hold, but as drivers that can steer their clinical trajectory.

Beyond expectations about treatments or surgical outcomes, self-efficacy represents a different but closely related form of expectancy—one directed not toward external interventions but toward the patient’s own capacity to manage symptoms and cope with challenges [20]. Rather than anticipating what a treatment will do for them, individuals with high self-efficacy hold expectations about what they can do themselves to influence their condition. This belief in personal agency has been shown to shape coping behaviors, persistence with activity, and adaptive responses to pain [21]. In musculoskeletal conditions, higher self-efficacy predicts lower pain severity, better physical functioning, and greater engagement in self-management behaviors [22]. In OA specifically, high self-efficacy has consistently been linked to better pain control, functional outcomes, and adherence to exercise in OA populations [23]. Thus, self-efficacy functions as an internal expectancy that may contribute to recovery by shaping coping responses and facilitating sustained engagement with meaningful activities.

Despite the growing recognition of their relevance, expectations remain inconsistently defined and measured in OA research. Studies vary in conceptual scope—defining recovery as pain reduction, restoration of function, return to work, or resumption of social participation—and employ heterogeneous instruments ranging from single-item global scales to domain-specific tools [24,25]. Moreover, most evidence arises from post-arthroplasty cohorts, with limited investigation in earlier disease stages, conservative care, or culturally diverse populations [26]. This fragmentation impedes synthesis and translation of findings into person-centered clinical practice.

Scoping reviews, guided by PRISMA-ScR methodology, are well-suited to address this challenge [27]. By mapping how expectations are conceptualized and linked to outcomes, this review aims to (i) clarify domains of patient expectations in OA; (ii) examine how expectations influence clinical outcomes and satisfaction with treatment; and (iii) identify gaps in evidence to inform future research directions.

## 2. Materials and Methods

This scoping review was conducted in accordance with the Preferred Reporting Items for Systematic Reviews and Meta-Analyses extension for Scoping Reviews (PRISMA-ScR) guidelines. The aim was to map and synthesize evidence regarding patient expectations and perceived recovery among individuals with osteoarthritis (OA).

### 2.1. Eligibility Criteria

We included studies that (i) investigated patients with knee, hip or generalized OA; (ii) examined expectations or self-efficacy; (iii) reported on clinical outcomes such as pain, function and treatment satisfaction; and (iv) were longitudinal, cross-sectional, qualitative studies or systematic reviews published in peer-reviewed journals. Case reports, editorials, expert opinions, and studies focusing exclusively on pharmacological efficacy without patient-centered outcomes were excluded.

### 2.2. Search Strategy

A comprehensive literature search was performed across PubMed, Embase, CINAHL, PsycINFO, and Web of Science from inception to 30 September 2025. Search terms combined concepts related to OA, expectations and self-efficacy. Boolean operators and indexing terms (MeSH, Emtree) were applied. The complete search strategy is available in Appendix A. Hand-searching of reference lists and gray literature was also performed.

### 2.3. Selection of Studies

Two reviewers independently screened titles, abstracts, and full texts. Disagreements were resolved through discussion or consultation with a third reviewer. A PRISMA flow diagram was created to document the selection process (Figure 1). 

### 2.4. Data Charting Process

Extracted data included author, year, study design, population, expectations assessed, and main findings (clinical outcomes). Extraction was performed independently by two reviewers and compared for accuracy. Extracted data can be found in Table 1.

**Figure 1 jcm-14-08440-f001:**
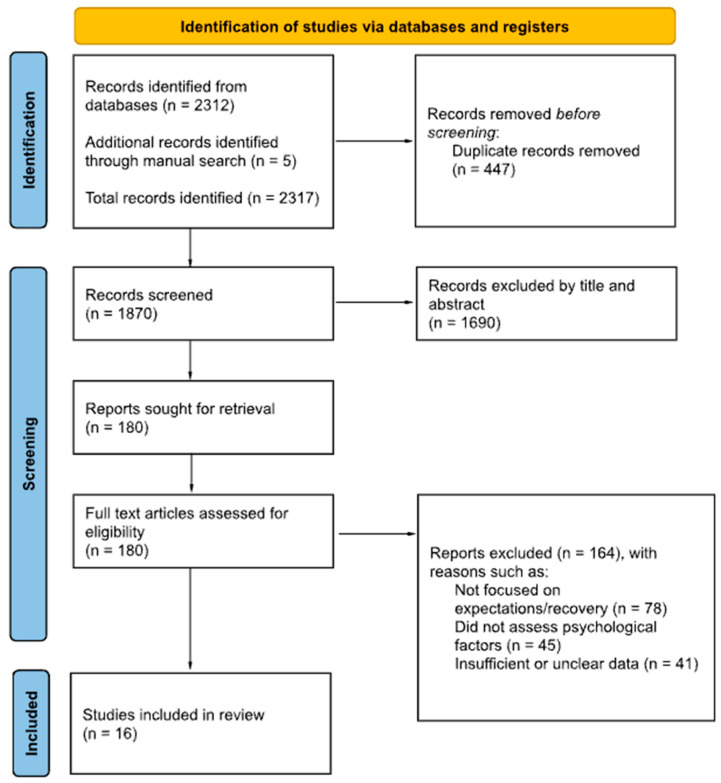
PRISMA Flow Diagram [37].

### 2.5. Synthesis of Results

Given heterogeneity, no meta-analysis was attempted, and findings were synthesized narratively. Evidence was summarized thematically across expectation domains and outcomes.

## 3. Results

### 3.1. Expectations About Surgery

Eleven studies examined the relationship between expectations about surgery and pain intensity, functional limitations and satisfaction. In five of them, participants underwent knee surgery, and in three of them, hip surgery.

Evidence on preoperative expectations indicates that patient expectations of improvement did not always align with clinical outcome. For instance, 85% expected to be pain-free, whereas only 43% achieved this outcome (*p* < 0.05); similarly, 52% expected to have no limitations in usual activities, but only 20% were actually free of such limitations postoperatively (*p* < 0.05) [28].

In Total Hip Arthroplasty (THA), 10–15% of patients remain dissatisfied due to unfulfilled expectations about the persistence of pain and ongoing functional limitations [9]. Patients who held expectations for improvements in nonessential activities reported lower rates of satisfaction [11]. More than 30% of patients had unfulfilled expectations regarding post-surgical improvement in walking long distances (31%), walking stairs (33%), and the ability to cut toenails (38%) [13]. Patients with better preoperative function scores were more likely to experience fulfillment of function-related expectations, except for daytime pain, transfer, exercise, and cutting toenails (0.01; *p* = 0.05) [31].

A similar pattern was found for Total Knee Arthroplasty (TKA). More than 40% of patients had unfulfilled expectations regarding walking middle long distances (up to 1.5 km) (40%), kneeling (44%), and squatting (47%) [13]. Among dissatisfied patients, 50% reported they were not as active as expected (OR = 0.17, *p* < 0.001), 32% reported being less active after surgery (OR = 0.35, *p* = 0.013), and 53% stated that their knee prevented them from engaging in desired activities (OR = 0.42, *p* = 0.016) [16].

Expectation fulfillment was consistently associated with improvements in pain and function at 6 and 12 months (β = 0.38, 95% CI: 0.35–0.40, *p* < 0.001), whereas baseline expectations were not [12]. Additionally, a higher risk of long-term dissatisfaction was observed in patients with high preoperative expectations regarding kneeling (RR 2.2, 95% CI: 0.9–5.5) and walking without aids (RR 2.4, 95% CI: 0.7–7.6) [29].

Expectations about the future resumption of household and social/recreational activities mediated the relationship between catastrophizing and follow-up pain severity (accounting for 12% of the variance) and follow-up physical function (accounting for 19% of the variance) [30]. Overall, positive patient expectations were associated with better postoperative pain and disability outcomes.

### 3.2. Expectations About Non-Surgical Management

Three studies examined the relationship between expectations regarding non-surgical treatments and pain intensity, functional limitations and satisfaction. Two of them were specifically focused on exercise-based interventions.

Overall, the evidence in conservative management contexts was mixed. There was no evidence of an association between patients’ baseline treatment expectations and pain reduction at 6 or 12 months after receiving non-surgical care. However, patients who received the treatment for which they held high expectations were nearly twice as likely to be classified as a treatment responder at 6 months (OR = 1.7, 1.06–2.79) and 12 months (OR = 1.9, 1.13–3.13) compared to those who did not state a preferred treatment [33].

Higher expectations toward exercise were associated with better adherence, particularly when patients perceived exercise as a credible, beneficial intervention. Bennell et al. reported that patients who expected high-intensity training to yield greater strength gains were more adherent to exercise programs [15]. Conversely, Wallis et al. found that low expectations toward non-surgical treatments were linked to reduced contact with clinicians and limited engagement with recommended strategies [4].

### 3.3. Expectations About Patients’ Ability to Cope or Self-Efficacy

Four studies examined the relationship between self-efficacy and various clinical and behavioral outcomes. In three of these studies, participants underwent surgery, while in one study, participants engaged in an exercise-based intervention.

Self-efficacy is associated with higher adherence to exercise programs, greater improvements in pain, physical activity levels and daily function in both hip and knee OA populations [15,34].

High self-efficacy for pain management at baseline resulted in reduced pain (37.43 ± 0.40, *p* < 0.01) and increased physical activity (5.05 ± 0.07; *p* < 0.01) at 3 and 12 months. High self-efficacy for management of other symptoms resulted in lower pain (35.78 ± 0.71, *p* < 0.01) and higher physical activity (5.08 ± 0.05, *p* < 0.01) at follow-ups [36].

Self-efficacy was also positively associated with quality of life. Wojcieszek et al. reported that a greater sense of self-competence was linked to better overall life quality in individuals with OA (α = 0.84, *p* < 0.001) [35].

## 4. Discussion

Across studies, expectation fulfillment emerged as the most consistent determinant of both satisfaction and functional improvement [12,13,29]. Positive and realistic expectations, particularly when combined with high self-efficacy, predicted favorable outcomes across pain and function domains. Overall, the findings of this review suggest that recovery in OA is a complex process in which patients’ expectations, and the degree to which they are fulfilled, play a central role. This aligns with broader evidence suggesting that recovery extends beyond pain reduction to include resumption of meaningful activities, maintenance of autonomy, and reconstruction of a sense of normality in daily life [9,38,39].

The association between initial expectations and clinical outcomes such as pain and physical function was inconsistent, especially after accounting for cofounders. This observation agrees with previous studies on knee arthroplasty, where preoperative expectations tend to lose predictive value once age, comorbidities, or baseline functional status are considered [12]. Nevertheless, some studies report that positive expectations are linked to better postoperative outcomes or greater improvements during exercise-based interventions [11]. Both high and low expectations may lead to satisfaction when fulfilled, as patients may adjust their preoperative expectations after surgery to reduce the imbalance between expectations and outcomes and thereby prevent dissatisfaction. Nevertheless, high expectations appear to offer an advantage over low expectations [32]. Expectations may therefore influence recovery indirectly by enhancing self-efficacy and emotional wellbeing, which in turn can promote motivation, adherence, and persistence during rehabilitation [40].

The most consistent finding in this review was the relationship between the fulfillment of expectations and treatment satisfaction. In both surgical and conservative settings, the alignment between what patients anticipate and what they ultimately experience reliably predicts satisfaction and perceived therapeutic success [13,16,29]. When outcomes fall short of expectations, frustration and dissatisfaction may arise, even when objective improvements occur. Conversely, alignment between expectations and outcomes strengthens patients’ sense of control and fosters a more positive appraisal of their recovery [41].

From a clinical standpoint, the gap between expectations and outcomes reflects not only individual perceptions but also the quality of the therapeutic relationship. Preoperative education and prehabilitation programs can enhance satisfaction by helping patients recalibrate their expectations and strengthen their self-efficacy before treatment begins [13,42]. These observations are compatible with the biopsychosocial model, which highlights the influence of psychological factors—such as self-efficacy, catastrophizing, and mood—on pain perception and recovery trajectories [43].

High self-efficacy has been linked to less pain, greater adherence, and better physical function [44,45]. In contrast, catastrophizing and depressive symptoms are associated with poorer functional outcomes and lower satisfaction, often mediating whether expectations are ultimately fulfilled [32]. Positive yet realistic expectations, especially when combined with high self-efficacy, may enhance perceived control and modulate pain through endogenous placebo mechanisms involving dopaminergic reward pathways [34,43].

Neurobiological evidence reinforces this view, showing that expectations act as physiological modulators of nociception. Positive expectations activate the descending pain inhibitory system via higher cortical regions such as the dorsolateral prefrontal cortex and rostral anterior cingulate cortex, which project to the periaqueductal gray and rostral ventromedial medulla and stimulate the release of endogenous opioids and dopamine, thereby inhibiting nociceptive transmission at the dorsal horn [44,46]. Activation of the mesolimbic reward system (nucleus accumbens) not only reduces pain but may also reinforce motivation for rehabilitation [45]. Conversely, negative expectations and catastrophizing may trigger nocebo mechanisms through cholecystokinin, which facilitates pain transmission and counteracts endogenous opioids, amplifying pain [47].

Altogether, these findings underscore the importance of routinely assessing patients’ expectations as part of clinical care. Recent literature has introduced the idea of “recovery as living well with pain,” which reframes recovery as the ability to live meaningfully despite ongoing symptoms. This concept emphasizes functionality and personal meaning over the complete disappearance of pain [48].

Clarifying what each individual hopes to achieve—whether pain relief, improved mobility, engagement in daily activities, or greater independence—can help clinicians tailor interventions and set shared, meaningful goals. When expectations are openly discussed and aligned with realistic outcomes, both satisfaction and adherence tend to improve [49,50].

Ultimately, these findings support a broader understanding of recovery in OA. Integrating patients’ expectations into therapeutic planning, and actively working to recalibrate them when needed, represents a cornerstone of contemporary physiotherapy and rehabilitation practice.

## 5. Conclusions

This scoping review shows that clinical outcome is strongly shaped by patients’ expectations, with expectation fulfillment emerging as one of the most consistent predictors of satisfaction, perceived treatment success, and functional improvement.

These findings underscore the importance of aligning clinical interventions with patient expectations. Incorporating systematic expectation assessment and recalibration into OA care may strengthen adherence, improve satisfaction, and support more meaningful functional progress.

Future work should examine expectations as modifiable therapeutic targets and identify how expectations interact with other variables such as symptom duration, extension and severity. Understanding these combinations may help identify which patient profiles benefit most from expectation-focused interventions. Overall, expectations emerge as a promising avenue for more person-centered physiotherapy and rehabilitation practice.

## Figures and Tables

**Table 1 jcm-14-08440-t001:** Extracted data.

Author (Year)	Design and Population	Expectations Assessed	Main Findings
Barker et al. (2023) [9]	Qualitative synthesis, HOA	Expectations about surgery (THA)	10–15% of patients remain dissatisfied due to unfulfilled expectations about the persistence of pain and ongoing functional limitations.
Mancuso et al. (1997) [11]	Prospective cohort, HOA	Expectations about surgery (THA)	Patients with expectations of improvements in nonessential activities reported lower rates of satisfaction.
Tilbury et al. (2016) [13]	Prospective cohort, HOA and KOA	Expectations about surgery	In THA, >30% of patients with unfulfilled expectations about “improvement in walking ability”: long distances” (31%), “walking stairs” (33%), and “improve ability to cut toenails” (38%). In TKA, >40% of patients with unfulfilled expectations about “walking middle long distances (up to 1.5 km’s)” (40%), “being able to kneel down” (44%) and “being able to squat” (47%).
Mannion et al. (2009) [28]	Prospective cohort, KOA	Expectations about surgery	Patients were overly optimistic about the likelihood of being pain-free (85% expected it, and 43% were pain-free; *p* < 0.05) and of not being limited in usual activities (52% expected it, and 20% experienced it; *p* < 0.05)
Noble et al. (2006) [16]	Prospective cohort, KOA	Expectations about surgery (TKA)	50% of dissatisfied patients reported they were not as active as they expected (OR = 0.17, *p* < 0.001). 32% of dissatisfied patients reported they were less active after surgery (OR = 0.35, *p* = 0.013). 53% of dissatisfied patients reported that their knee kept them from doing activities they wanted to do (OR = 0.42, *p* = 0.016).
Lim et al. (2025) [12]	Prospective cohort, KOA	Expectations about surgery (TKA)	Expectation fulfillment was consistently associated with improvements in pain and function at 6 and 12 months (ß: 0.38, 95% CI: 0.35 to 0.40, *p* < 0.001) but baseline patient expectations were not.
Yapp et al. (2020) [29]	Prospective cohort, KOA	Expectations about surgery (TKA)	Higher risk of long-term dissatisfaction in patients with high preoperative expectations of kneeling (RR 2.2, 95% CI 0.9–5.5) and walking without aids (RR 2.4, 95% CI 0.7–7.6).
Sullivan et al. (2011) [30]	Prospective cohort, KOA	Expectations about surgery (TKA)	Expectations about the future resumption of household and social/recreational activities mediated the relation between catastrophizing and follow-up pain severity (accounted for 12% of the variance); and follow-up physical function (accounted for 19% of the variance).
Mancuso et al. (2009) [31]	Prospective cohort, HOA	Expectations about surgery (THA)	Patients with better preoperative scores were more likely to have fulfillment of function-related expectations (with the exceptions of daytime pain, transfer, exercise and cut toenails) (0.01; *p* = 0.05)
Hafkamp et al. (2020) [32]	Systematic review, HOA and KOA	Expectations about surgery	Both high and low expectations could lead to satisfaction when expectations are fulfilled, as patients might change their preoperative expectations postoperatively in order to diminish imbalance between expectations and outcomes to prevent dissatisfaction. However, high expectations have an advantage over low expectations.
Wallis et al. (2019) [4]	Qualitative review, KOA	Expectations about treatment	People with low expectations of treatment ended up having limited contact with health professionals.
Foster et al. (2010) [33]	Prospective cohort, KOA	Expectations about treatment (exercise or acupuncture)	No evidence of a relationship between patients’ baseline treatment expectations and pain reduction at 6 or 12 months. Patients who received the treatment for which they had high expectations were almost twice as likely to be classified as a treatment responder at 6 months (OR = 1.7, 1.06, 2.79) and 12 months (OR = 1.9, 1.13, 3.13).
Hawker et al. (2006) [34]	Review, KOA	Expectations about surgery (TKA) and self-efficacy	Positive patient expectations related to better postoperative pain and disability. Self-efficacy is associated with greater improvements in pain and disability.
Bennell et al. (2011) [15]	Review, HOA and KOA	Expectations about exercise and self-efficacy	Higher expectations about exercise related to better adherence, with high-intensity training expected to result in greater strength gains. Higher self-efficacy is also associated with higher adherence and better pain and function outcomes.
Wojcieszek et al. (2023) [35]	Cross-sectional, KOA	Self-efficacy	A higher sense of self-competence was associated with better quality of life (α = 0.84, *p* < 0.001).
Degerstedt et al. (2020) [36]	Prospective cohort, HOA and KOA	Self-efficacy	High self-efficacy for pain management at baseline resulted in reduced pain (37.43 ± 0.40, *p* < 0.01) and increased physical activity (5.05 ± 0.07; *p* < 0.01) at the follow-ups; High self-efficacy for management of other symptoms resulted in lower pain (35.78 ± 0.71, *p* < 0.01) and higher physical activity (5.08 ± 0.05, *p* < 0.01) at 3 and 12 months.

Abbreviations. α, Cronbach’s alpha; CI, confidence interval; HOA, hip osteoarthritis; KOA, knee osteoarthritis; OR, odds ratio; RR, relative risk; THA, total hip arthroplasty; TKA, total knee arthroplasty.

## Data Availability

The original contributions presented in this study are included in the article/Appendix A. Further inquiries can be directed to the corresponding author.

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
