# Peer review of "The Role of Patient Expectations in Treatment Outcome and Satisfaction in Osteoarthritis: A Scoping and Mapping Review"

_jcm, 2025, doi:10.3390/jcm14238440_

Round 1

Reviewer 1 Report

Comments and Suggestions for Authors

Dear Authors, Thank you for the opportunity to review your manuscript. The topic is timely and clinically meaningful, and the paper is generally well written. I have listed several comments and suggestions below that I believe will help improve the clarity, consistency, and overall quality of the manuscript.

- The abbreviations in lines 53 and 200 should be defined in full. Please also review other abbreviations throughout the manuscript.

- In Figure 1, adjust the alignment of the text and lines for consistency.

- After line 281, there is an extra space between the reference number and the period (.). Please correct this formatting issue.

- In the Discussion section, add relevant references supporting the neurobiological mechanisms of expectations and expand the corresponding content.

- In the Conclusion, multiple topics are presented within a single paragraph, which may reduce readability. It is recommended to shorten the content or divide it into separate paragraphs.

Author Response

Reviewer 1

Comment 1: The abbreviations in lines 53 and 200 should be defined in full. Please also review other abbreviations throughout the manuscript.

Response 1: We thank the reviewer for this helpful observation. All abbreviations at the indicated lines have now been fully defined, and we have carefully reviewed the entire manuscript to ensure consistent and appropriate abbreviation use.

Comment 2: In Figure 1, adjust the alignment of the text and lines for consistency.

Response 2: We appreciate the reviewer’s attention to detail. The alignment of text and lines in Figure 1 has now been corrected to ensure full consistency and clarity.

Comment 3: After line 281, there is an extra space between the reference number and the period (.). Please correct this formatting issue.

Response 3: Thank you for noting this formatting issue. The spacing error has been corrected as requested.

Comment 4: In the Discussion section, add relevant references supporting the neurobiological mechanisms of expectations and expand the corresponding content.

Response 4: We thank the reviewer for this valuable suggestion. We have now expanded the relevant section in the Discussion and incorporated additional references addressing the neurobiological mechanisms underlying expectations.

Comment 5: In the Conclusion, multiple topics are presented within a single paragraph, which may reduce readability. It is recommended to shorten the content or divide it into separate paragraphs.

Response 5: We appreciate the reviewer’s recommendation. The Conclusion has been revised to improve readability by reorganizing and shortening the content, resulting in clearer and more coherent final paragraphs.

Reviewer 2 Report

Comments and Suggestions for Authors

Thank you for inviting me to review this paper. This scoping review compiles studies linking OA “recovery” recognition to parameters beyond pain reduction—such as expectations and forgotten scores—and emphasizes the importance of expectation management in achieving high post-treatment satisfaction.

My major concern is that the Results section is presented in a subjective, conclusion-oriented manner that does not meet scientific reporting standards. The Results should adopt a data-centric style, even if the included studies are heterogeneous; these details could be summarized in the Results or a Supplementary table. For example, the authors should present that “the expectation–satisfaction correlations using a/b/c scores presented very strong correlations (r>0.8, p<0.01) [Chan Y. Biostatistics 104: correlational analysis. 2003]” in the Results, and only then described subjectively as “strong and consistent” in the Discussion or Conclusion, rather than as currently stated in the Results (page 6, line 211).

Other comments:

  1. Page 2, line 56: The manuscript states that clinicians typically rely on “symptom-focused, prioritizing radiographic outcomes or pain intensity scales” to define OA recovery; however, page 5, line 200 indicates that numerous surveys and scores play vital roles in OA recovery assessment. Please reconcile this.
  2. Page 3, line 116: The databases searched were MEDLINE, Embase, CINAHL, PsycINFO, and Web of Science. Please clarify why PubMed was not searched, given its central role in biomedical literature and its indexing of MEDLINE records.
  3. Page 4, Figure 1: Please redesign. The dark purple background reduces legibility; side bars should use vertically aligned text (not one letter per line); and arrows should be consistently aligned.
  4. Page 4, line 194: It is unclear why a “source” is provided for the PRISMA guideline here. Please clarify.
  5. Page 5, line 201: Comparative, interpretive statements currently in the Results should be moved to the Discussion.

Author Response

Reviewer 2

Comment 1: My major concern is that the Results section is presented in a subjective, conclusion-oriented manner that does not meet scientific reporting standards. The Results should adopt a data-centric style, even if the included studies are heterogeneous; these details could be summarized in the Results or a Supplementary table. For example, the authors should present that “the expectation–satisfaction correlations using a/b/c scores presented very strong correlations (r>0.8, p<0.01) [Chan Y. Biostatistics 104: correlational analysis. 2003]” in the Results, and only then described subjectively as “strong and consistent” in the Discussion or Conclusion, rather than as currently stated in the Results (page 6, line 211).

Response 1: We thank the reviewer for this important observation. In response, we have expanded Table 1 to clearly present all relevant data from the included studies, as suggested. We also rewrote and reorganised the Results section to adopt a fully data-centric structure, removing interpretative wording and ensuring clearer, more transparent reporting. The interpretative elements that were previously in the Results have now been moved to the Discussion, where they are appropriately contextualised. We believe these changes substantially improve the clarity and scientific rigour of the manuscript.

Comment 2: Page 2, line 56: The manuscript states that clinicians typically rely on “symptom-focused, prioritizing radiographic outcomes or pain intensity scales” to define OA recovery; however, page 5, line 200 indicates that numerous surveys and scores play vital roles in OA recovery assessment. Please reconcile this.

Response 2: We thank the reviewer for highlighting this point. We have clarified the distinction in the revised manuscript. The sentence on page 2 refers to the common clinical practice, in which recovery is often interpreted narrowly and primarily assessed through radiographic findings or pain intensity. In contrast, the section on page 5 refers to validated research instruments—such as WOMAC, KOOS, HOOS, or the Oxford Knee Score—that provide a multidimensional evaluation of OA outcomes, including function, symptoms and quality of life.

Comment 3: Page 3, line 116: The databases searched were MEDLINE, Embase, CINAHL, PsycINFO, and Web of Science. Please clarify why PubMed was not searched, given its central role in biomedical literature and its indexing of MEDLINE records.

Response 3: We thank the reviewer for bringing this to our attention. This was an inadvertent error in the manuscript. We did not search MEDLINE directly; instead, we conducted our search through PubMed, which provides full access to MEDLINE-indexed records. The text has been corrected to reflect that our searches were performed in PubMed, Embase, CINAHL, PsycINFO, and Web of Science.

Comment 4: Page 4, Figure 1: Please redesign. The dark purple background reduces legibility; side bars should use vertically aligned text (not one letter per line); and arrows should be consistently aligned.

Response 4: We appreciate the reviewer’s attention to detail. The alignment of text and lines in Figure 1 has now been corrected to ensure full consistency and clarity.

Comment 5: Page 4, line 194: It is unclear why a “source” is provided for the PRISMA guideline here. Please clarify.

Response 5: We recognise that it may have created confusion. We have now removed the “source” reference from that sentence and clarified the text to indicate that the review follows the PRISMA-ScR guidelines.

Comment 6: Page 5, line 201: Comparative, interpretive statements currently in the Results should be moved to the Discussion.

Response 6: Subjective descriptors have been removed from the Results and are now restricted to the Discussion section. We believe these revisions substantially improve scientific rigor and clarity.